# Uncovering *PheCLE1* and *PheCLE10* Promoting Root Development Based on Genome-Wide Analysis

**DOI:** 10.3390/ijms25137190

**Published:** 2024-06-29

**Authors:** Changhong Mu, Wenlong Cheng, Hui Fang, Ruiman Geng, Jutang Jiang, Zhanchao Cheng, Jian Gao

**Affiliations:** Key Laboratory of National Forestry and Grassland Administration/Beijing for Bamboo & Rattan Science and Technology, International Center for Bamboo and Rattan, State Forestry and Grassland Administration, Beijing 100102, China; muchanghong@icbr.ac.cn (C.M.); chengwenlong@icbr.ac.cn (W.C.); fanghui@icbr.ac.cn (H.F.); gengruiman@126.com (R.G.); jiangjutang@icbr.ac.cn (J.J.)

**Keywords:** Moso bamboo, *CLE* gene family, apical tissue development, *PheCLE1*, *PheCLE10*

## Abstract

Moso bamboo (*Phyllostachys edulis*), renowned for its rapid growth, is attributed to the dynamic changes in its apical meristem. The CLAVATA3/EMBRYO SURROUNDING REGION-RELATED (CLE) family genes are known to play crucial roles in regulating meristem and organ formation in model plants, but their functions in Moso bamboo remain unclear. Here, we conducted a genome-wide identification of the *CLE* gene family of Moso bamboo and investigated their gene structure, chromosomal localization, evolutionary relationships, and expression patterns. A total of 11 *PheCLE* genes were identified, all of which contained a conserved CLE peptide core functional motif (Motif 1) at their C-termini. Based on Arabidopsis classification criteria, these genes were predominantly distributed in Groups A–C. Collinearity analysis unveiled significant synteny among *CLE* genes in Moso bamboo, rice, and maize, implying potential functional conservation during monocot evolution. Transcriptomic analysis showed significant expression of these genes in the apical tissues of Moso bamboo, including root tips, shoot tips, rhizome buds, and flower buds. Particularly, single-cell transcriptomic data and in situ hybridization further corroborated the heightened expression of *PheCLE1* and *PheCLE10* in the apical tissue of basal roots. Additionally, the overexpression of *PheCLE1* and *PheCLE10* in rice markedly promoted root growth. PheCLE1 and PheCLE10 were both located on the cell membrane. Furthermore, the upstream transcription factors NAC9 and NAC6 exhibited binding affinity toward the promoters of *PheCLE1* and *PheCLE10*, thereby facilitating their transcriptional activation. In summary, this study not only systematically identified the *CLE* gene family in Moso bamboo for the first time but also emphasized their central roles in apical tissue development. This provides a valuable theoretical foundation for the further exploration of functional peptides and their signaling regulatory networks in bamboo species.

## 1. Introduction

There are approximately 50 million hectares of bamboo forests in the world, which are primarily located in the tropical and subtropical regions of Africa, Asia, and Central and South America, encompassing around 1642 bamboo species [1,2]. Bamboo, as a renewable resource, is extensively used in various fields including paper production, construction materials, and household commodities [3]. Bamboo forests, through their rapid growth and biomass accumulation, significantly absorb and store carbon dioxide, serving as a vital carbon sink resource in combating global climate change [4]. In 2021, the global export value of bamboo products reached USD 3.599 billion, with China’s exports accounting for 71.1%, totaling USD 2.56 billion [1]. China’s bamboo forest area has reached 7.5627 million hectares, representing 3.3% of the total forest area [1]. Moso bamboo is the most widely cultivated species in China. The renewal and expansion of Moso bamboo forests primarily rely on an asexual propagation system consisting of mother bamboo, rhizomes, and new bamboo shoots. The establishment, maintenance, and differentiation of apical meristematic tissues such as rhizome tips, shoot tips, and root tips are fundamental to the asexual reproduction of Moso bamboo. Additionally, Moso bamboo is renowned for its astonishing growth rate. It can rapidly grow up to 20 m within 45 to 60 days with a peak growth rate of up to 1 m per day [5]. This exceptional characteristic primarily stems from the development of its apical and intercalary meristematic tissues [6]. Despite some progress in anatomical studies concerning apical tissues in bamboos, research on the genetic regulatory mechanisms governing these tissues remains limited.

Plant development, growth, and environmental responses are significantly influenced by the critical roles of signaling peptides and hormones in facilitating communication between cells. The CLE family, comprising the signaling peptides CLAVATA3 (CLV3) and Embryo Surrounding Region (ESR), is a significant class of signaling peptides in plants [7,8]. The CLE family genes generally encode small peptides ranging from 50 to 200 amino acids, which include a small N-terminal secretion signal peptide and a conserved C-terminal CLE motif consisting of 14 amino acids with a variable region in between the two components [9]. A few CLE precursor peptides contain multiple CLE motifs. Mature CLE peptides undergo post-translational modifications, such as hydroxylation and glycosylation before cleavage [10,11]. These peptides play crucial roles in regulating the development of apical meristems, the formation of vascular tissues, lateral root growth, and fruit development. To date, a total of 1628 CLE peptides have been predicted across the genomes of 57 plant species [12], including Arabidopsis [13], rice [14], maize [15], poplar [16], tomato [17], and soybean [18]. However, studies on CLE peptides of Moso bamboo remain unreported. A comprehensive investigation into the *CLE* gene family in Moso bamboo is essential for understanding the genetic regulatory mechanisms governing the development of its apical meristematic tissues.

Although extensive studies have been conducted on the *CLE* gene family in model plants such as Arabidopsis and rice, only a few *CLE* genes have been functionally characterized. In Arabidopsis, *CLV3* is a key member of the CLE family. In the shoot apical meristem, the transcription factor WUSCHEL (WUS) maintains the stem cell state by activating *CLV3* expression [19]. Subsequently, CLV3 peptides inhibit the activity of *WUS* through their recognition by receptors such as CLAVATA1 (CLV1), CLAVATA2 (CLV2)/CORYNE (CRN), and RECEPTOR-LIKE PROTEIN KINASE 2 (RPK2) [20,21,22]. This precise feedback regulation mechanism between *WUS* and *CLV3* serves as the basis for regulating stem cell homeostasis in the shoot apical meristem. In addition, SHOOT MERISTEMLESS (STM), another transcription factor, can bind to the promoter region of the *CLV3* gene and enhance WUS binding to the *CLV3* gene promoter region through direct interaction with WUS. The heterodimerization of WUS and STM, as well as their simultaneous binding to two sites on the *CLV3* promoter, are necessary for regulating *CLV3* gene expression and ensuring robust maintenance of stem cells [23,24]. Moreover, in root apical tissues, *CLE40* is expressed in the differentiated root cap and inhibits the expression of *WUSCHEL RELATED HOMEOBOX 5* (*WOX5*) by interacting with receptor-like kinases ARABIDOPSIS CRINKLY 4 (ACR4) and CLV1 expressed in columella stem cells [25]. This regulatory mechanism effectively controls both the proliferation and differentiation of stem cells in the root apical meristem. Mutations in *cle40* lead to a reduction in primary roots and a decrease in root apical meristem tissue [26]. In studies of model plants such as Arabidopsis and poplar, scientists have discovered that *CLE41* and *CLE44*, specifically expressed in the phloem, significantly promote the proliferation of vascular cambium stem cells. Plants lacking these peptides show a notable reduction in the number of procambial cells, especially when both CLE41 and CLE44 are absent, which amplifies this effect [27]. CLE42 is functionally similar to CLE41/44. The *CLE41/44* genes, encoding the TRACHEARY DIFFERENTIATION INHIBITORY FACTOR (TDIF) peptide, bind to the PHLOEM INTERCALATED WITH XYLEM/TDIF RECEPTOR (PXY/TDR) receptor to regulate cell division in the vascular cambium [28]. Additionally, the TDIF–PXY signaling pathway can regulate lateral root occurrence through BIN2-mediated auxin signaling [29]. In addition to these classical regulatory mechanisms, CLE peptides also participate in hormone and stress signal responses. For instance, CLE25 peptides respond to drought signals by transmitting signals to the leaves through BARELY ANY MERISTEM (BAM) receptors and synergistically working with ABSCISIC ACID (ABA) to regulate stomatal closure to reduce water loss [30]. Furthermore, *OsCLE48* responds to auxin signals, thereby regulating the development of the apical meristematic tissue in Arabidopsis [31]. *CLE6* enhances the effect of CLE41/TDIF in promoting procambial proliferation, which is intensified by the exogenous application of the auxin analog NAPHTHALENEACETIC ACID (NAA) and diminished in the presence of the auxin polar transport inhibitor NAPHTHYLPHTHALAMIC ACID (NPA) [32]. In summary, CLE family genes play a crucial role in plant growth and development, particularly in maintaining the activity of meristematic tissues and organ morphogenesis. Therefore, delving into the study of the *CLE* gene family in Moso bamboo is vital for elucidating the genetic regulatory mechanisms behind the development and maintenance of its apical meristems.

To understand the *PheCLE* gene family in Moso bamboo, this study integrates insights from Arabidopsis, rice, and maize. We meticulously described the structural features of these genes and systematically classified them according to the established functional categories from Arabidopsis. Through collinearity analysis with the *CLE* genes in rice and maize, the phylogenetic relationships among them have been revealed. Additionally, this study explored the tissue-specific expression of *PheCLE* genes in Moso bamboo. Furthermore, the overexpression of *PheCLE1* and *PheCLE10* in rice demonstrated their crucial regulatory roles in plant root development. The regulatory networks of *PheCLE1* and *PheCLE10* in the development of Moso bamboo root apical tissue have been preliminarily elucidated. These findings not only enrich our understanding of the Moso bamboo *CLE* gene family but also provide a solid theoretical foundation for further research on the functional mechanisms of CLE peptides in apical tissue development in bamboo plants.

## 2. Results

### 2.1. Identification of Members of the CLE Family and Analysis of Protein Characteristics in Moso Bamboo

In order to identify and characterize the *CLE* gene family in Moso bamboo, 11 *CLE* genes were screened from the Moso bamboo genome database based on the polypeptide sequences of the *CLE* gene family in Arabidopsis and rice [33]. These genes were renamed according to their positions on the chromosomes, namely *PheCLE1* to *PheCLE11* (Figure 1). The distribution pattern of *CLE* gene family members on 10 chromosomes (Chr3, Chr4, Chr7, Chr8, Chr9, Chr13, Chr15, Chr16, Chr17, and Chr24) of Moso bamboo was revealed using TBtools software (v2.096) [34]. The results showed that these genes were relatively evenly distributed on the chromosomes with the exception of two genes (*PheCLE7* and *PheCLE8*) located on Chr15. The lengths of 11 PheCLE proteins ranged from 78 to 157 amino acids, and four of them (PheCLE1, PheCLE7, PheCLE10, and PheCLE11) had a signal peptide at the N-terminus. The highest relative molecular weight of these proteins was 16.3 KDa (PheCLE6), and the lowest was 8.3 KDa (PheCLE9). The isoelectric points of all the PheCLE proteins, except for PheCLE11, were greater than 7, indicating that the peptides in this family were basic. In addition, the 11 PheCLE proteins were identified as hydrophilic liposoluble and unstable proteins (Appendix A).

### 2.2. Analysis of PheCLE Gene Structure and Conserved Motifs

To investigate the structural diversity and sequence characteristics of the *PheCLE* genes in Moso bamboo, as well as to enhance our comprehension of their functional roles and evolutionary significance within the *CLE* gene family, we analyzed the gene structures and conserved motifs of the *CLE* genes. The analysis of gene structure revealed that *PheCLE* genes in the Moso bamboo exhibited varying exon numbers: *PheCLE1* and *PheCLE10* displayed a single exon, while *PheCLE8* and *PheCLE11* possessed three exons with the remaining genes containing two exons. The sequence features of the 11 Moso bamboo CLE proteins were analyzed using the MEME online tool, revealing the presence of six conserved motifs. All PheCLE protein sequences contained a core functional motif (Motif 1) located at the C-terminus (Appendix A) with the consensus sequence SKRRVPNGPDPIHN (Figure 2). The gene structures of the Moso bamboo CLE family members show diversity, suggesting potential functional variations among these proteins. The consistency of the core functional motif emphasizes its central role in regulating plant growth and development. 

### 2.3. Evolutionary Analysis of CLE Genes

To explore the evolutionary relationships among *CLE* gene family members across different species, a phylogenetic tree was constructed using the maximum likelihood method (ML) based on the protein sequences of Arabidopsis, rice, maize, and Moso bamboo. There are 32 CLEs in Arabidopsis, 39 in rice, 28 in maize, and 11 in Moso bamboo (Figure 3, Appendix A). Referring to the classification of the CLE family in Arabidopsis [35], the CLEs from the four species were divided into four subfamilies. In particular, Group A primarily consisted of seven members from Moso bamboo. Group B comprised three members while Group C had one member; no distribution of Moso bamboo’s CLEs was observed in Group D. Additionally, the *PheCLE* genes showed a closer phylogenetic relationship with the *CLE* gene families of rice and maize.

### 2.4. Collinearity Analysis of CLE Genes

The expansion and contraction of gene families during evolution can influence gene structure, thereby affecting gene functional differentiation. Therefore, we conducted an analysis of the genomic collinearity of the *CLE* gene family in Moso bamboo and the collinearity maps between different species. A collinearity analysis was conducted for CLE family members in Moso bamboo as well as those in rice and maize (Figure 4). In the Moso bamboo genome, there were two pairs of *PheCLE* genes distributed on different chromosomes, which may be due to gene duplication events potentially resulting from segmental duplication on different chromosomes. Comparative syntenic maps were constructed at the whole-genome level between Moso bamboo and both rice and maize. Eight *CLE* genes in Moso bamboo showed homology with nine *CLE* genes in rice, while ten *CLE* genes in Moso bamboo exhibited homology with eleven *CLE* genes in maize. These findings indicate that the *CLE* gene family in Moso bamboo has not only experienced gene duplication events within the species, leading to their distribution on different chromosomes but also maintained a certain degree of homology across species. Comparisons with rice and maize reveal a high degree of conservation and clear collinearity, highlighting the importance and potential functional conservation of these genes in the evolution of monocots.

### 2.5. Analysis of Cis-Acting Elements in the PheCLE Gene Promoters

To explore the regulatory factors and mechanisms of *PheCLE* gene expression, we further analyzed the promoter regions of the *CLE* gene family in Moso bamboo. The analysis of the promoter region located 2000 bp upstream of the start codon of the *PheCLE* gene family using PlantCARE (Figure 5) revealed that the promoters of these genes predominantly contained elements responsive to plant hormones such as methyl jasmonate, abscisic acid, gibberellin, and auxin as well as environmental factors including light and cold. Additionally, these promoter regions contained a multitude of binding sites for transcription factors such as MYB, NAC, and WOX. Specifically, the *PheCLE* gene family exhibited a high abundance of hormone-responsive elements, particularly those responsive to methyl jasmonate, totaling 64, with *PheCLE11* containing 12 of these elements in its promoter region. In terms of environmental factor response elements, light-responsive elements were the most prevalent with a total of 66, and all CLE family members possessed them. The promoters of *PheCLE* genes universally contained binding sites for the transcription factors MYB, NAC, and WOX, with *PheCLE6* having 17 MYB binding sites, *PheCLE9* having five WOX binding sites, and *PheCLE1* having six NAC binding sites. Moreover, out of 11 *PheCLE* genes, five exhibited promoter regions containing elements associated with meristem expression. Overall, these findings suggest that the involvement of *PheCLE* genes in bamboo meristematic tissue development is regulated by plant hormones, light, and various transcription factors.

### 2.6. Expression Patterns of CLE Family Genes in Moso Bamboo

To gain a comprehensive understanding of the functional roles of CLE family genes across various developmental stages and tissue types in Moso bamboo, we utilized publicly available transcriptome datasets to construct an expression atlas for the bamboo CLE family genes (Figure 6A–D) [6,33,36,37]. Our analyses revealed phase-specific high expression levels of *PheCLE* gene family members at the apex of bamboo shoots, particularly during rapid elongation phases. Specifically, *PheCLE1*, *PheCLE2*, and *PheCLE10* exhibited pronounced expression at the apex of 100 cm in height. *PheCLE3* and *PheCLE6* were highly expressed at the apex of 300 cm in height. The expressions of *PheCLE5*, *PheCLE7*, and *PheCLE8* were predominantly localized at the apical region of shoots measuring 600 cm in height (Figure 6A). Moreover, the CLE family genes demonstrated diverse expression patterns across different tissue types in Moso bamboo with significant expressions in roots, shoots, and rhizome buds but negligible levels detected in leaves (Figure 6B). In the floral organs, CLE family genes were predominantly expressed in flower buds (Figure 6C). Notably, single-cell transcriptome data from the apical tissues of Moso bamboo basal roots further indicated that except for *PheCLE2* expressed in fundamental tissues, other family members were specifically expressed in root cap cells (Figure 6D). In summary, the expression of CLE family genes in bamboo was primarily concentrated in various apical tissues with vigorous division ability. The observed expression profiles of the *CLE* genes in Moso bamboo align with their critical roles in maintaining meristematic tissues and influencing organ morphogenesis in model plants, suggesting a similar important role in the development of bamboo meristematic tissues. Moreover, the distinct expression profiles of these genes at different developmental phases and across various tissue types in Moso bamboo offer valuable clues for future detailed research into their unique roles in bamboo growth and development.

### 2.7. In Situ Hybridization Analysis of PheCLE1 and PheCLE10

Through comprehensive analysis of the research results in the previous section, we have identified that *PheCLE10* was highly expressed in the rhizome roots of Moso bamboo. Additionally, both *PheCLE1* and *PheCLE10* genes were significantly expressed in Moso bamboo basal root caps. This finding suggested that *PheCLE1* and *PheCLE10* might play important regulatory roles in the development of Moso bamboo roots. To further elucidate the expression locations of these two genes, we conducted in situ hybridization experiments in the basal root tips (Figure 7A–H). Analysis of cross-sections and longitudinal sections of the basal root tip revealed clear hybridization signals for *PheCLE1* and *PheCLE10* in the root tip region, further confirming their specific expression in root tip cells. Consequently, it can be inferred that these genes potentially play pivotal roles in the growth and development of Moso bamboo root tips. 

### 2.8. Overexpression of PheCLE1 and PheCLE10 Leads to Rice Root Elongation

Subsequently, this study focused on the molecular functions of *PheCLE1* and *PheCLE10*. Considering the low genetic transformation efficiency in bamboo, coupled with the phylogenetic analyses previously discussed, further functional studies were conducted in rice. The overexpression of *PheCLE1* and *PheCLE10* in rice resulted in a significant enhancement of total root length in 7-day-old seedlings compared to wild-type (WT) plants (Figure 8A). This result indicates that *PheCLE1* and *PheCLE10* play a role in positively regulating root development in rice. Furthermore, the subcellular localization of *PheCLE1* and *PheCLE10* was also investigated in this study. The coding sequences of these two genes were fused with the pCAMBIA2300-35S-eGFP vector to construct fusion expression vectors. The transient expression in tobacco leaves was observed using laser confocal microscopy, revealing the localization of both PheCLE1 and PheCLE10 on the cell membrane (Figure 8B). The findings not only enhance our understanding of the roles of *PheCLE1* and *PheCLE10* in Moso bamboo root development but also provide important clues for further exploring the functions of CLE peptides in plant root architecture and functional regulation.

### 2.9. Screening for Upstream Regulatory Factors of PheCLE1 and PheCLE10 Genes

To advance insights into the regulatory mechanisms by *PheCLE1* and *PheCLE10* over apical tissue development in Moso bamboo, this study conducted an extensive investigation of the upstream regulatory factors of these two genes. Specifically, we successfully cloned the promoter sequences of *PheCLE1* and *PheCLE10* and inserted them into the pHIS2 vector for subsequent yeast one-hybrid library screening experiments. The results indicated that the NAC transcription factors, PheNAC9 and PheNAC6, specifically interacted with the promoter regions of *PheCLE1* and *PheCLE10*, respectively, suggesting that these transcription factors might play roles in regulating the expression of the *PheCLE* genes (Figure 9). Furthermore, a dual-luciferase reporter assay system was employed to verify the effects of PheNAC9 and PheNAC6 on the promoter activities of *PheCLE* genes. The experimental outcomes confirmed that PheNAC9 specifically bound to the promoter region of *PheCLE1*, while PheNAC6 displayed binding specificity to the promoter region of *PheCLE10*, and both significantly enhanced the transcriptional activities of their respective promoters (Figure 10). Additionally, the visualization of single-cell transcriptomic data revealed that both PheNAC9 and PheNAC6 were significantly expressed in Moso bamboo basal root tips (Appendix A). These results indicated that PheNAC9 and PheNAC6 might be involved in apical tissues development in Moso bamboo root by directly regulating the expression of *PheCLE1* and *PheCLE10*. Overall, the findings of this study provide a new perspective on the function of *PheCLE* genes in root apical tissues of Moso bamboo. Additionally, the research uncovers the potential regulatory functions of NAC transcription factors within this developmental process. This lays a solid foundation for in-depth studies on the molecular mechanisms underlying the development of apical tissues in bamboo and other plants.

## 3. Discussion

Moso bamboo, as an important non-timber forest resource in China, has significant economic, ecological, and cultural value. Regardless of the life cycle of bamboo or the high growth of bamboo stems, the apical meristem plays a crucial role in the morphogenesis and growth development of bamboo. Currently, research on the apical meristem of bamboo plants mainly focuses on anatomical aspects, while studies on the molecular regulatory mechanisms underlying its development are relatively lacking. CLE family genes play an important role in plant growth and development, particularly in the regulation of apical meristems [38]. While extensive research has been conducted on the *CLE* gene family in model plants such as Arabidopsis and rice, studies focusing on bamboo remain relatively scarce. This study identified and analyzed the *CLE* gene family in Moso bamboo, revealing its chromosomal distribution patterns, gene structure, conserved motifs, and evolutionary characteristics of the family. The identification of 11 CLE family genes in Moso bamboo represents a relatively modest number compared to that observed in other model plants. This may be attributed to gene loss or gene fusion events during the long evolutionary history of Moso bamboo. Moreover, the uneven distribution of these genes across chromosomes could potentially arise from processes such as gene duplication, transposition, or chromosomal rearrangements. The hydrophilicity and instability of the PheCLE protein family members suggest their potential roles in rapid regulation and signal transduction within plants. The alkaline characteristics of these proteins and the presence of signaling peptides may suggest that they play a specific role in the transmission of extracellular signals. Additionally, the structural diversity of the *PheCLE* gene family may contribute significantly to their functional diversity during bamboo growth and development. In particular, the consistency of the CLE core functional motifs suggests that these proteins may possess central biological functions in the growth regulation of bamboo.

Following Arabidopsis CLE family classification guidelines [35], *CLE* genes were divided into four subfamilies. In Moso bamboo, *PheCLE* genes were primarily distributed in Groups A–C, while no distribution of *PheCLE* genes was observed in Group D. In Arabidopsis, Group D *CLE* genes, including *CLE41*, *CLE42*, and *CLE44*, maintained vascular tissue homeostasis by inhibiting xylem differentiation and promoting the proliferation of vascular cambium cells [39]. The absence of corresponding *CLE* genes in Moso bamboo suggests that it may have evolved different mechanisms to regulate its vascular tissue development during evolution. These alternative mechanisms may involve other *CLE* gene family members or different signaling pathways to adapt to its unique growth and developmental requirements. Additionally, there could be unidentified structural variants of Moso bamboo genes that differ from Group D *CLE* genes but potentially possess similar functions involved in bamboo vascular tissue development. Furthermore, evolutionary analysis revealed a closer phylogenetic relationship between the *PheCLE* genes and the *CLE* genes of rice and maize. Collinearity analysis among Moso bamboo, rice, and maize indicated some degree of functional conservation among monocot *CLE* gene family members during evolution, highlighting the potential existence of similar biological functions across different species.

Although it has been established that the *CLE* gene family is crucial for the regulation of plant shoot and root apical meristems, vascular tissues, and fruit development [40], research on its functions in the monocot plant Moso bamboo remains insufficient. In this study, transcriptomic sequencing data were utilized to visually analyze the expression patterns of *PheCLE* genes at both tissue and single-cell levels in Moso bamboo. The results revealed distinct expression patterns of *PheCLE* genes across different developmental stages, various tissues, and diverse cell clusters in Moso bamboo. Notably, the expression of *PheCLE* genes markedly escalated in the shoot tips during accelerated growth stages of bamboo shoots. Additionally, substantial expression of the Moso bamboo *CLE* gene family members was also observed in roots, bamboo shoots, rhizome buds, and floral buds. Overall, the majority of *PheCLE* genes exhibited increased expression in the apical tissues of Moso bamboo, indicating their crucial influence on the regulation of growth and development in these apical tissues by potentially influencing cellular differentiation and proliferation for organ formation. These results broaden our understanding of the *CLE* gene family and open new avenues for investigating their roles in the growth and development of monocot plants.

Through predictive analysis, we have identified that both CLE1 and CLE10 in Moso bamboo were secretory signaling peptides. Single-cell transcriptome analysis has revealed the distinct expression of these peptides in the root cap cells of Moso bamboo basal root apical tissue, which is similar to the expression pattern of *CLE40* in Arabidopsis [41]. *CLE40*, expressed in differentiated root cap cells, was perceived by the plasma membrane receptor kinase ACR4, thereby inhibiting the expression of *WOX5* in the quiescent center [42]. In the distal meristem of the root, the CLE40–ACR4–WOX5 signaling pathway is formed, which regulates the proliferation and differentiation of root apical stem cells. Mutations in the *CLE40* and *ACR4* genes resulted in a reduction in primary root length and an increase in root tip stem cells [26,43]. Our study demonstrated that the overexpression of *PheCLE1* and *PheCLE10* in rice led to significant root growth, suggesting that the CLE signaling peptides in Moso bamboo may regulate the development of root apical meristem through a similar signaling pathway, thereby promoting root elongation. Additionally, *CLE45* also plays a crucial role in root development. In Arabidopsis, *CLE45* regulated root development through its receptor BAM3 [44]. Treatment with CLE45 peptides significantly reduced the size of the proximal meristem and increased the number of lateral roots [45]. This provides a new perspective for studying the roles of *PheCLE1* and *PheCLE10* in the root development of Moso bamboo. Our research has revealed that *PheCLE1* and *PheCLE10* exhibit a closer phylogenetic relationship with *CLE45*, indicating their potential involvement in regulating similar biological processes in Moso bamboo. This suggests their influence on the homeostasis of the root apical meristem and the formation of lateral roots. In summary, PheCLE1 and PheCLE10 may interact with specific receptor kinases in Moso bamboo, forming a complex regulatory network that finely tunes root development. In the future, it is necessary to further explore the signaling pathways and gene functions of these two genes in the root apical meristem of bamboo plants. This will help to gain a more comprehensive understanding of the regulatory mechanisms underlying root development in bamboo plants. Moreover, this study investigated the upstream regulatory mechanisms of the *PheCLE* genes in Moso bamboo. It was revealed that the *PheCLE* genes may be influenced by various plant hormones (methyl jasmonate, abscisic acid, gibberellin, and auxin), environmental stimuli (light exposure and low temperatures), and transcription factors (MYB, NAC, and WOX), and participate in the development of bamboo meristem tissue. These findings are consistent with the function of *CLE* genes in maintaining stem cell homeostasis and response to stress and hormone signals in model plants [46,47]. Furthermore, yeast one-hybrid library screening and dual-luciferase reporter assays indicated that NAC9 and NAC6 can directly upregulate the expression of *PheCLE1* and *PheCLE10*. Similar feedback regulation mechanisms involving *CLE* genes have been observed in other plants such as Arabidopsis and *Marchantia polymorpha*. The NAC transcription factor XVP activated *CLE44* gene expression in the maintenance of vascular tissue stem cells, whereas the TDIF peptide produced inhibited *XVP* expression [48]. A comparable mechanism involving MpJIN-CLE22 was observed in the development of the apical meristematic tissues of the gametophyte stems in *Marchantia polymorpha* [49]. Thus, comprehensive future investigations are essential to elucidate the regulatory mechanism between *PheCLE1*, *PheCLE10*, and NAC transcription factors in Moso bamboo. Overall, this research not only highlights the pivotal roles of *PheCLE1* and *PheCLE10* in the development of root apical tissues in Moso bamboo but also demonstrates their intricate interactions with NAC transcription factors, offering invaluable clues for a deeper understanding of the regulatory mechanisms governing root apical tissues in bamboo plants.

## 4. Materials and Methods

### 4.1. Experimental Samples and Cultivation Conditions

Moso bamboo basal root tissues were collected from the natural distribution area in Guangde City, Anhui Province, China (119.41769 E, 30.89371 N). Experimental samples were taken from the basal root apical tissues (~2 cm) of approximately 50 ± 2 cm high Moso bamboo spring shoots and were used in subsequent in situ hybridization experiments. 

The *PheCLE1* and *PheCLE10* genes were seamlessly cloned into the pC1300 vector, which was then transformed into *Agrobacterium tumefaciens* GV3101. This construct was subsequently introduced into wild-type rice (Zhonghua 11) using the *Agrobacterium*-mediated transformation method. Stable overexpression lines of *PheCLE1* and *PheCLE10* were isolated through selection with G418 [50]. Phenotypic observations were conducted on seven-day-old wild-type (WT) and transgenosis (transgenic homozygous lines) rice. The total root length of both wild-type and transgenic rice was quantified using IMAGE J software (v1.54g).

### 4.2. Identification and Protein Characterization of CLE Family Members of Moso Bamboo

The study utilized bioinformatics techniques at the protein and nucleotide levels to pinpoint members of the Moso bamboo CLE family. Initially, CDS and protein data of the CLE family members from model plants (Arabidopsis, rice, and maize) were sourced from several databases: Tair (https://www.arabidopsis.org/, accessed on 23 January 2024), China Rice Data Center (https://www.ricedata.cn/, accessed on 23 January 2024), and Phytozome (https://phytozome-next.jgi.doe.gov/, accessed on 23 January 2024). These sequences served as references for conducting a BLASTN search within the Moso bamboo genome database, with a stringency threshold set at an e-value of 10^−5^, to preliminarily screen for potential Moso bamboo *CLE* genes. Furthermore, a Hidden Markov Model (HMM) was constructed using the protein sequences of Arabidopsis, rice, and maize. HMMER 3.0 software was utilized to search within the Moso bamboo protein database. By integrating the results from these two analytical methods, the list of candidate genes was further refined. To further validate these results, the protein sequences were aligned using MEGA11.0 software. Based on the functional peptide segments of CLE, gene sequences that did not conform to the structural characteristics of CLE were eliminated, thus pinpointing the accurate sequence of the *CLE* genes in Moso bamboo. Additionally, the detection of signal peptides within Moso bamboo CLE proteins was carried out using SignalP 5.0 (https://services.healthtech.dtu.dk/services/SignalP-5.0/, accessed on 20 March 2024).

The protein sequences of Moso bamboo CLE peptides as determined previously were input into the TBtools software (v2.096). Utilizing the Protein Parameter Calc program, we calculated the physicochemical properties of the peptides.

### 4.3. Structural and Motif Analysis of PheCLE Genes

TBtools software was employed to illustrate the gene structures of *PheCLE*, while their conserved motifs were visually analyzed through the MEME online platform (http://memesuite.org/tools/meme, accessed on 2 April 2024).

### 4.4. Evolutionary Analysis of CLE Genes

Multiple sequence alignments of CLE protein sequences from Moso bamboo, Arabidopsis, rice, and maize were performed using ClusterW. Subsequently, the *CLE* gene family’s phylogeny was deduced through Maximum Likelihood analysis in the MEGA11.0 environment, setting the bootstrap parameter to 1000. Visualization enhancements were applied via the iTOL platform (http://itol.embl.de, accessed on 10 April 2024).

### 4.5. Collinearity Analysis of CLE Genes

To investigate the conservation of CLE family genes on different chromosomes in Moso bamboo and other species, collinearity analysis was performed on CLE family members from bamboo, rice, and maize using MCScanX software (v1.3.1) [51]. The results of the collinearity analysis were visualized using the Advanced Circos program within TBtools software.

### 4.6. Promoter Analysis of PheCLE Genes

Genomic data from Moso bamboo were extracted using TBtools software to obtain the sequences 2000 bp upstream of the *PheCLE* genes start site, which were defined as promoter regions. Subsequently, cis-acting elements within these promoters were predicted using the PlantCARE (http://bioinformatics.psb.ugent.be/webtools/plantcare/html/, accessed on 15 April 2024). Finally, visualization of the predicted data was achieved through the ‘Gene Structure View (Advanced)’ in TBtools.

### 4.7. Gene Expression Pattern Analysis of PheCLE Genes

To further explore the regulatory mechanism of the *PheCLE* genes, exploratory analysis was conducted on the expression trends of *PheCLE* genes using transcriptome data from various developmental stages of Moso bamboo shoots, different floral organs of Moso bamboo, various tissue organs of Moso bamboo, as well as single-cell transcriptome data from the apical tissue of Moso bamboo basal root. Additionally, heatmaps were generated using TBtools software.

### 4.8. Subcellular Localization of PheCLE Proteins

Transient expression studies were initiated by integrating the coding sequences of *PheCLE1* and *PheCLE10* into the pCAMBIA2300-35S-eGFP vector. Subsequently, these constructs were transformed into the *Agrobacterium tumefaciens* strain GV3101. The activated *Agrobacterium* strains were collected and resuspended in tobacco injection solution (10 mM MES, pH 5.6, 10 mM MgCl_2_, 20 μM acetosyringone) to adjust the bacterial suspension to an OD value of 0.8–1.0. The resuspension was left at room temperature for 2–4 h. Then, using a 1 mL syringe, the suspension was injected into the abaxial side of 4-week-old tobacco leaves. The injected tobacco plants were incubated in the dark for 24 h [52]. The eGFP fluorescence was visualized under a Zeiss laser confocal microscope (Zeiss, Oberkochen, Germany). Primers for the fusion constructs are detailed in Appendix A.

### 4.9. In Situ Hybridization

The specific DNA probes for the *PheCLE1* and *PheCLE10* genes were designed and synthesized by GENWIZ with detailed information provided in Appendix A. Fresh Moso bamboo root tip tissues (~2 cm) were washed and fixed in 4% paraformaldehyde solution. Subsequently, the samples underwent dehydration, clearing, wax infiltration, and embedding processes. Paraffin-embedded samples were sectioned into 8 µm slices using a microtome. Following pretreatment of the tissue sections, we performed pre-hybridization, hybridization, and immunodetection. The detailed protocol for in situ hybridization follows the procedures described by Cheng et al. [36,53]. We used anti-digoxigenin antibodies for labeling and detected the signals with NBT/BCIP solution. Images were recorded using a Zeiss microscope in brightfield mode.

### 4.10. Yeast One-Hybrid (Y1H) Assay

We cloned the promoter sequences of *PheCLE1* and *PheCLE10* and subsequently attached these sequences to the pHIS2 vector using a seamless cloning method with DNA Assembly Mix Plus (LABLEAD, Beijing, China). Following the protocol described by Li [50], we utilized a yeast one-hybrid library from Moso bamboo shoots to identify proteins binding to the promoter of *PheCLE1* and *PheCLE10*. The coding sequences of *PheNAC9* and *PheNAC6*, identified through screening, were inserted into the pGADT7 vector for further validation. The two sets of plasmids, pGADT7-PheNAC9 and pHIS2-PheCLE1, as well as pGADT7-PheNAC6 and pHIS2-PheCLE10, were each transformed into Y187 yeast cells. These transformed cells were incubated on SD/-His-Leu-Trp + X-α-gal solid medium at 30 °C for approximately three days.

### 4.11. Dual Luciferase Assay

The *PheNAC9* and *PheNAC6* coding sequences were separately transferred into the pGreenII-62-SK vector, while the previously cloned promoter sequences of *PheCLE1* and *PheCLE10* were each linked to the pGreenII-0800-LUC vector. The relationship between the effector and reporter was assessed with the E1910 Dual-Luciferase Reporter Assay System (Promega, Madison, WI, USA). Additionally, the activities of LUC and REN were measured using a GloMax^®^ 20/20 luminometer. The primer sequences employed in these experiments are detailed in Appendix A.

## 5. Conclusions

This study identified 11 *PheCLE* genes and revealed that each PheCLE protein sequence contained a crucial functional motif at the C-terminus. *CLE* genes in Moso bamboo, rice, and maize showed a high level of conservation and obvious collinearity, revealing the importance and potential functional conservation of these genes in the evolution of monocots. In addition, research has revealed that the *PheCLE* genes exhibited significantly expressed in the apical tissues of Moso bamboo, including roots, bamboo shoots, rhizome buds, and flower buds. The promoter regions of *PheCLE* genes were enriched in elements responsive to plant hormones and environmental factors as well as numerous binding sites for transcription factors such as MYB, NAC, and WOX. Yeast one-hybrid and dual-luciferase assays demonstrated that NAC9 and NAC6 distinctly interacted with the promoter of *PheCLE1* and *PheCLE10*, respectively, thereby initiating their expression. Combined with single-cell transcriptomic data and in situ hybridization experiments, it was revealed that *PheCLE1* and *PheCLE10* exhibited specific expression in apical tissue of Moso bamboo basal roots. The overexpression of these genes significantly promoted the growth of rice roots, suggesting their crucial regulatory roles in plant root development. In summary, this study revealed the molecular characteristics and evolutionary conservation of members of the Moso bamboo *CLE* gene family while emphasizing their potential importance in the development of apical tissues, which provide a theoretical basis for studying the regulatory mechanisms governing monocot apical tissue growth.

## Figures and Tables

**Figure 1 ijms-25-07190-f001:**
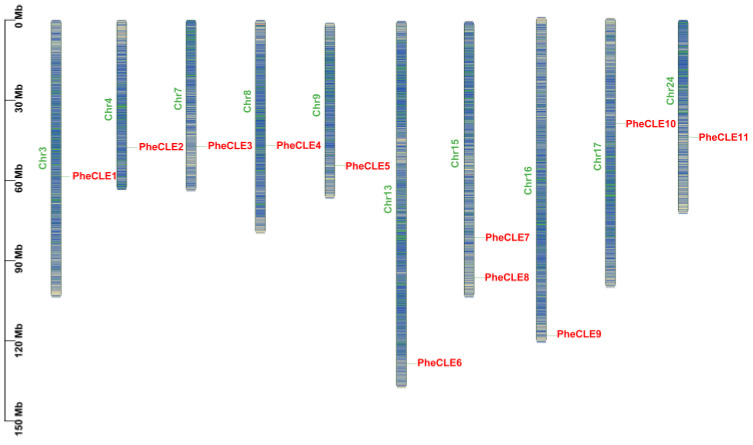
Chromosome locations of CLE gene family members of Moso bamboo. Striped boxes represent chromosomes containing gene density information. The scale represents the length of the Moso bamboo chromosomes. Red font represents members of the bamboo *CLE* gene family, and green font represents Moso bamboo chromosomes. Y-axis: Chromosome length (Megabases, Mb). X-axis: Chromosome numbers.

**Figure 2 ijms-25-07190-f002:**
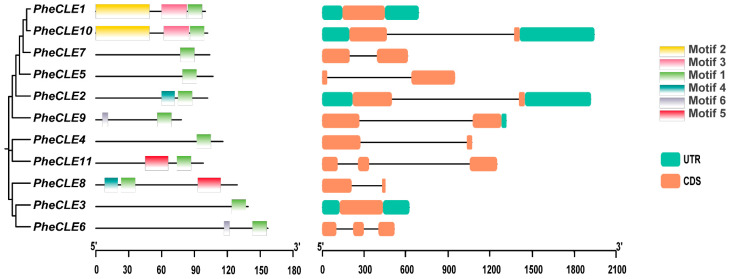
Gene structure and conserved motif analysis of *CLE* gene family members in Moso bamboo. The left figure represents the phylogenetic and conserved motif analysis of the *CLE* family with different colored squares representing different conserved motifs. The right figure represents the gene structure of *CLE* family members, where the blue square represents the UTR region, the orange square represents the exon, and the black line represents the intron. The left axis: Conserved Motif Length (amino acids, aa). The right axis: Gene Structure Length (base pairs, bp).

**Figure 3 ijms-25-07190-f003:**
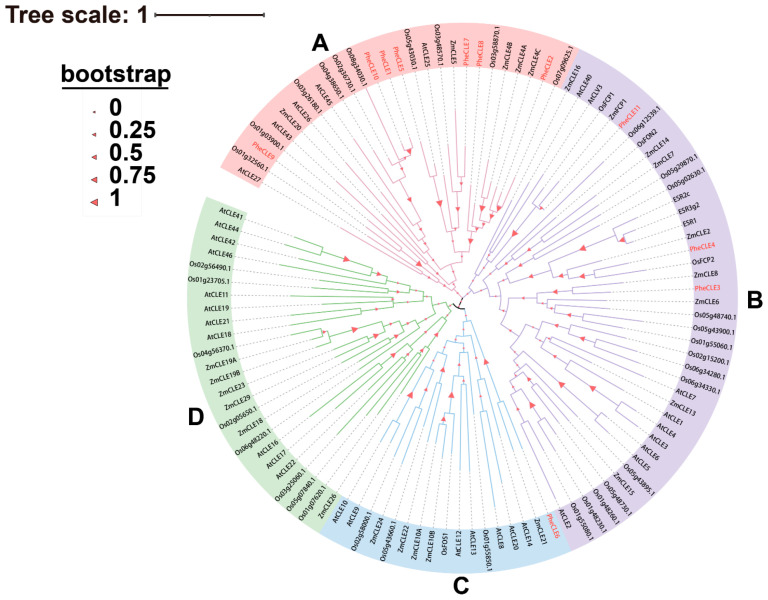
Evolutionary analysis of the *CLE* gene family. The 110 CLE protein sequences from Arabidopsis, rice, maize, and Moso bamboo can be divided into four subgroups: A, B, C, and D. The red font represents the Moso bamboo CLEs, and bootstrap values are represented by triangles of different sizes.

**Figure 4 ijms-25-07190-f004:**
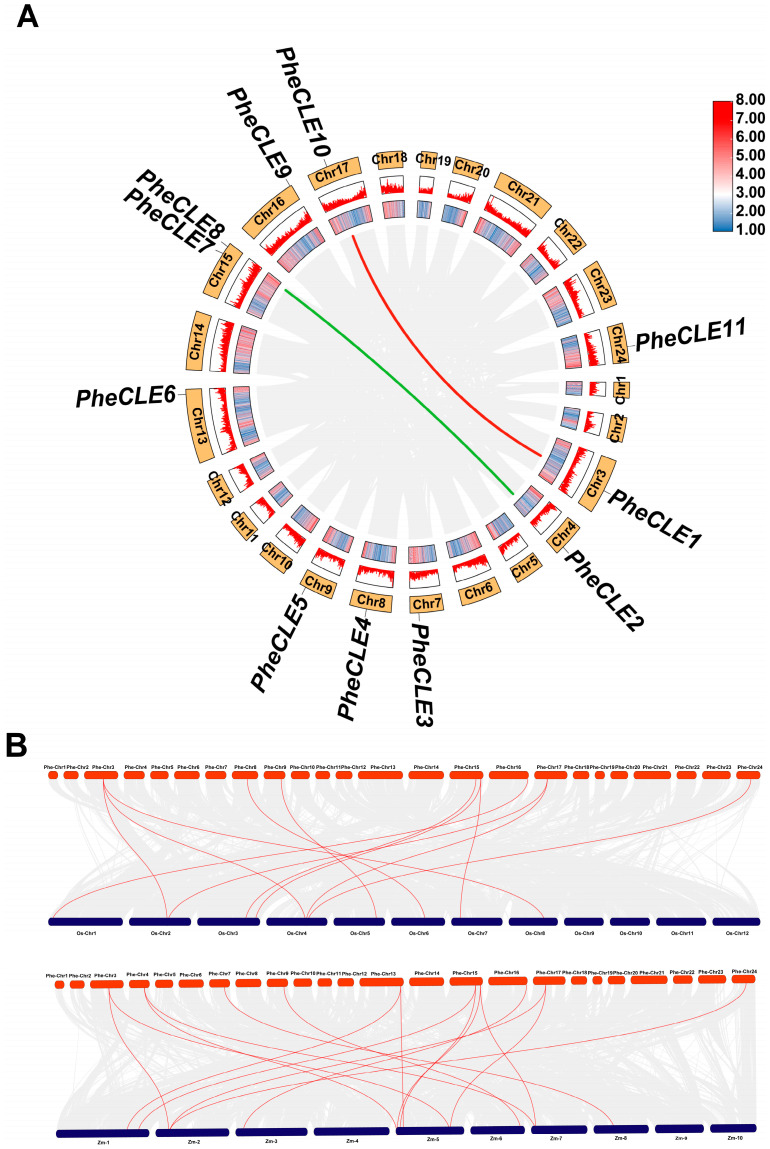
Collinearity analysis of *CLE* gene family members. (**A**) In the collinearity region of CLE family members within Moso bamboo, the red and green lines indicate the duplication relationship between two pairs of *PheCLE* genes. (**B**) The collinear relationship between Moso bamboo, rice, and maize CLE members. The relationship of duplicated genes between *PheCLEs*, *OsCLEs*, and *ZmCLEs* was indicated with red lines.

**Figure 5 ijms-25-07190-f005:**
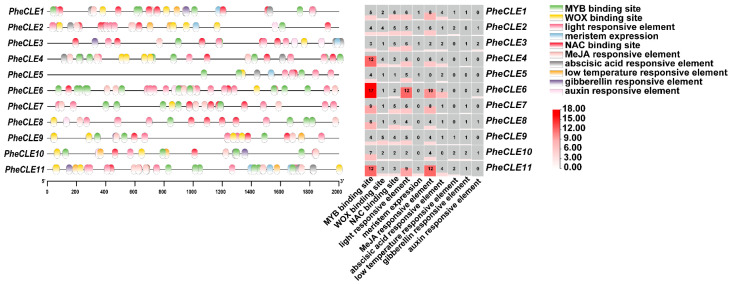
Analysis of promoter regions in the *PheCLE* genes. The blocks of different colors represent different cis-acting elements, and the heatmap represents the number of cis-acting elements for each *PheCLE* gene. The darker the color, the more components there are.

**Figure 6 ijms-25-07190-f006:**
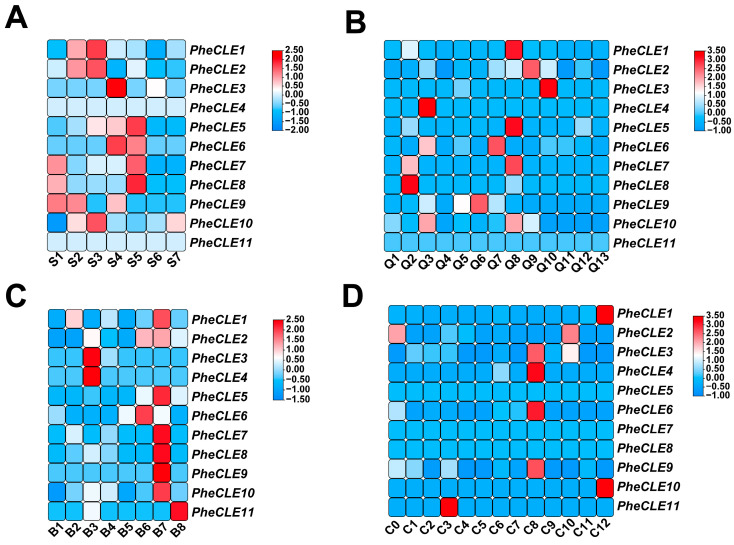
Analysis of *PheCLE* gene expression patterns based on transcriptome data. (**A**) Expression levels of *PheCLE* genes in different developmental stages of Moso bamboo shoots. S1: winter bamboo shoot tip, S2: 50 cm spring bamboo shoot tip, S3: 100 cm spring bamboo shoot tip, S4: 300 cm spring bamboo shoot tip, S5: 600 cm spring bamboo shoot tip, S6: 900 cm spring bamboo shoot tip, S7: 1200 cm spring bamboo shoot tip. (**B**) Expression levels of *PheCLE* genes in different tissue types of Moso bamboo. Q1: Rhizome, Q2: Rhizome bud, Q3: Rhizome root, Q4: 0.1 cm root on bamboo shoot, Q5: 0.5 cm root on bamboo shoot, Q6: 2 cm root on bamboo shoot, Q7: 10 cm root on bamboo shoot, Q8: 300 cm bamboo shoot tip, Q9: 300 cm bamboo shoot in the middle, Q10: 300 cm bamboo shoot base, Q11: Leaf, S12: Leaf sheath, S13: Culm sheath. (**C**) Expression levels of *PheCLE* genes in different floral organs of bamboo. B1: Leaves, B2: Pistils, B3: Stamens, B4: Young embryos, B5: Glumes, B6: Lemma, B7: Flower bud, B8: Bracts. (**D**) Expression profiles of *PheCLE* genes across different cell groups in the basal root apical tissue of Moso bamboo. Specifically, C0, C1, C5, and C10 corresponded to ground tissues, C2 to transition cells, C6 to the epidermis, C7, C8, C11, and C12 to the root cap, C9 to initial cells, and C3 and C4 to undefined tissues.

**Figure 7 ijms-25-07190-f007:**
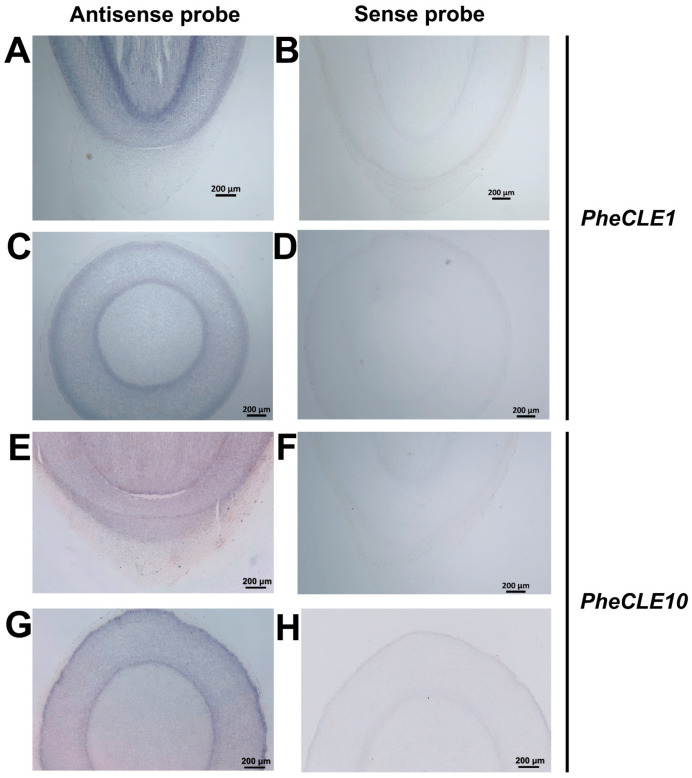
In situ hybridization experiment of *PheCLE1* and *PheCLE10* in the basal root tips of Moso bamboo. Longitudinal (**A**,**B**,**E**,**F**) and cross (**C**,**D**,**G**,**H**) sections of the root tip tissue of Moso bamboo basal root. In addition, (**A**,**C**,**E**,**G**) are the hybridization results of the target gene antisense probe, while (**B**,**D**,**F**,**H**) are the hybridization results of the target gene sense probe. Bar = 200 µm.

**Figure 8 ijms-25-07190-f008:**
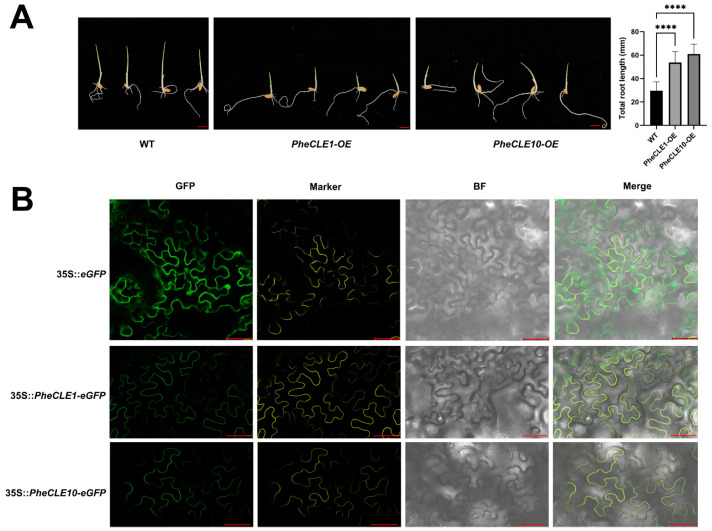
Functional analysis of *PheCLE1* and *PheCLE10* genes. (**A**) Phenotypic analysis of WT and *PheCLE1* and *PheCLE10* transgenic lines rice seeds after 7 days of germination. Bar = 10 mm. Quantification of total root length in WT, *PheCLE1-OE*, and *PheCLE10-OE* lines. Data are means ± SE (*n* = 15 roots). Asterisks indicate a statistically significant difference between WT and transgenic plants (*t*-test, **** *p* < 0.0001). (**B**) Subcellular localization analysis of PheCLE1 and PheCLE10 in tobacco leaves. Vectors carrying 35S::PheCLE1-eGFP and 35S::PheCLE10-eGFP were transiently transformed into tobacco leaves, using 35S::eGFP as the positive control and YFP as the cell membrane marker. Images of N. benthamiana leaves were captured 48 h post-infiltration using confocal microscopy. Bar = 50 µm.

**Figure 9 ijms-25-07190-f009:**
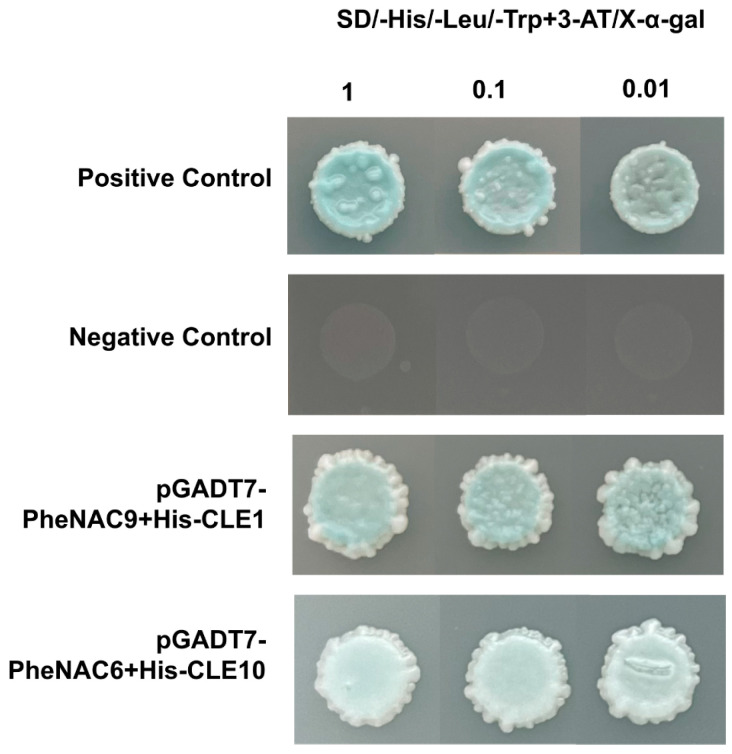
*PheCLE1* and *PheCLE10* genes yeast one-hybrid results. pHis2-53+pGADT7-53 was used for positive reference, and pHis2-53+pGADT7-Rec2 was used for negative reference. Growth patterns of the transformants at dilutions of 1, 0.1, and 0.01 on SD/-His-Leu-Trp plates.

**Figure 10 ijms-25-07190-f010:**
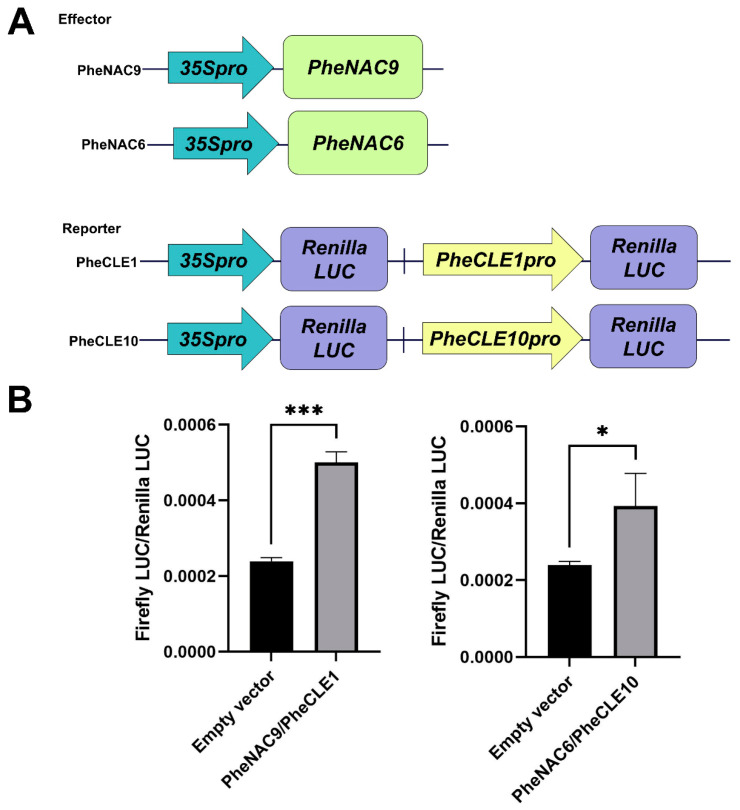
*PheCLE1* and *PheCLE10* genes dual luciferase experiment results. (**A**) Schematic representations of the reporter and effector constructs. (**B**) The luminescence ratio of firefly LUC to Renilla LUC. Results are expressed as means ± standard error for three replicates. Statistical significance compared to the control with empty vector is denoted by asterisks (*t*-test, * *p* < 0.05, *** *p* < 0.001).

## Data Availability

All the data to support the study results in this paper are in the Appendix A.

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
