# Peer review of "Uncovering *PheCLE1* and *PheCLE10* Promoting Root Development Based on Genome-Wide Analysis"

_ijms, 2024, doi:10.3390/ijms25137190_

Round 1
Reviewer 1 Report
Comments and Suggestions for Authors
In this study, we identified 11 PheCLE genes of the CLE gene family of Moso bamboo and described their gene structure, chromosomal localization, evolutionary relationships, and expression patterns.
Overall, the paper is descriptive, although various analyses are given, but they are presented very superficially and without justification.
Such works are quite abundant and follow a certain model. However, there should be a rationale for studying a particular gene family as some initial stage of a larger work. This study is done to report.
Specific data, gene sequences, and an alignment of these genes are needed to analyze their relatedness. In this case, I don't see the relatedness between these genes; just having Motif 1 at their C-termini does not guarantee their relatedness. Studying the DNA sequences of homologs in this genome and for related species is also necessary.
This is a rather complicated study if done on an all-encompassing level.
Each step needs to be justified, e.g., why these studies were necessary:
257 In situ hybridization analysis of PheCLE1 and PheCLE10.
In this chapter, it is unclear what was done, how, and what antibodies were used. In the methods section, also no protocol.
There is no description of the protocols and no justification for them. Overall, the work is rather descriptive and virtual. The appended practical results are rather included to add completeness to this study.
Reviewer 2 Report
Comments and Suggestions for Authors
This manuscript, entitled 'Uncovering PheCLE1 and PheCLE10 Promoting Root Development Based on Genome-Wide Analysis' by Mu et al., presents a comprehensive study of the CLE gene family in Moso bamboo, particularly focusing on the roles of PheCLE1 and PheCLE10 in root development. The research involves genome-wide identification, structural analysis, chromosomal localisation, evolutionary relationships, and expression patterns of CLE genes in Moso bamboo. Moreover, it explores the molecular functions of PheCLE1 and PheCLE10, showing their significant expression in the apical tissues and their role in enhancing root growth through overexpression experiments in rice. This study provides an extensive genome-wide analysis of the CLE gene family in Moso bamboo, addressing a gap in the existing literature and offering a detailed overview of gene structures, conserved motifs, and chromosomal localisation. The functional analysis of PheCLE1 and PheCLE10, including overexpression experiments in rice, effectively demonstrates their roles in enhancing root development, thus providing valuable insights into their biological functions. The investigation into upstream regulatory factors, especially the interaction of NAC transcription factors with the PheCLE1 and PheCLE10 promoters, provides a deeper understanding of the regulatory networks involved in root development. The aims of this study are clear, and the results are of interest to me. I have no major comments but would like to offer some suggestions below:
While the study identifies key regulatory factors, the mechanistic insights into how precisely PheCLE1 and PheCLE10 influence root development remain somewhat superficial. More detailed mechanistic studies, including downstream signalling pathways and target genes, would provide a more comprehensive understanding. Further discussion or a future plan would be beneficial to address this point.
In the manuscript, several complex sentences make the manuscript challenging to read. Please combine related ideas, remove redundancy, and revise to enhance readability. For example:
- L52-L54 "The CLE (CLAVATA3/Embryo surrounding region-related) family constitutes a significant class of signalling peptides in plants, referred to as CLAVATA3 (CLV3) and Embryo Surrounding Region (ESR), respectively." should be simplified to "The CLE family, comprising the signalling peptides CLAVATA3 (CLV3) and Embryo Surrounding Region (ESR), is a significant class of signalling peptides in plants."
- L105-L109 "This study conducted a comprehensive investigation of the Moso bamboo genome, focusing on the PheCLE gene family, based on existing knowledge from Arabidopsis, rice, and maize. The structural features of bamboo CLE genes were meticulously described, and a systematic classification was performed according to functional categorisations derived from Arabidopsis." should be revised to "To understand the PheCLE gene family in Moso bamboo, this study integrates insights from Arabidopsis, rice, and maize. We meticulously described the structural features of these genes and systematically classified them according to the established functional categories from Arabidopsis, highlighting their potential roles in cellular communication and development."
Additionally, the manuscript would benefit from being polished by a native English-speaking biologist.
Comments on the Quality of English LanguageIn the manuscript, several complex sentences make the manuscript challenging to read. Please combine related ideas, remove redundancy, and revise to enhance readability. For example:
- L52-L54 "The CLE (CLAVATA3/Embryo surrounding region-related) family constitutes a significant class of signalling peptides in plants, referred to as CLAVATA3 (CLV3) and Embryo Surrounding Region (ESR), respectively." should be simplified to "The CLE family, comprising the signalling peptides CLAVATA3 (CLV3) and Embryo Surrounding Region (ESR), is a significant class of signalling peptides in plants."
- L105-L109 "This study conducted a comprehensive investigation of the Moso bamboo genome, focusing on the PheCLE gene family, based on existing knowledge from Arabidopsis, rice, and maize. The structural features of bamboo CLE genes were meticulously described, and a systematic classification was performed according to functional categorisations derived from Arabidopsis." should be revised to "To understand the PheCLE gene family in Moso bamboo, this study integrates insights from Arabidopsis, rice, and maize. We meticulously described the structural features of these genes and systematically classified them according to the established functional categories from Arabidopsis, highlighting their potential roles in cellular communication and development."
Additionally, the manuscript would benefit from being polished by a native English-speaking biologist.
Reviewer 3 Report
Comments and Suggestions for Authors
The study by Changhong Mu and colleagues provides a comprehensive genome-wide analysis of the CLE gene family in Moso bamboo (Phyllostachys edulis), focusing on the identification and functional analysis of PheCLE1 and PheCLE10. Knowledge of molecular mechanisms of root development and their potential regulatory mechanisms is an important topic of molecular biology. The authors successfully identified 11 CLE genes in Moso bamboo, providing detailed insights into their chromosomal localization, gene structure, and evolutionary relationships. The study combined in silico analysis with experimentally validating the roles of PheCLE1 and PheCLE10 in root development.
Overexpression experiments demonstrated that PheCLE1 and PheCLE10 significantly promote root growth, corroborating the bioinformatics findings. Overall, this study makes significant contributions to the understanding of the CLE gene family in Moso bamboo, particularly PheCLE1 and PheCLE10. The combination of genome-wide analysis, experimental validation, and exploration of regulatory mechanisms provides a robust framework for future research.
Minor remarks:
- line 128 “revealed using TBtools software[34]”. Reference 34. Bai, Y.; Dou, Y.; Xie, Y.; Zheng, H.; Gao, J., Phylogeny, transcriptional profile, and auxin-induced phosphorylation modification 630 characteristics of conserved PIN proteins in Moso bamboo (Phyllostachys edulis). International Journal of Biological Macromolecules 631 2023, 234-247 does not apply TBtoos.
- lines 436 – 437 “several databases: Tair (https://www.arabidopsis.org/), China Rice Data Center (https://www.ricedata.cn/), and Phytozome (https://phytozome-next.jgi.doe.gov/).
When these databases were accessed?
- line 461 “using MEGA7 software” and line 461 “MEGA11.0 environment”. Why two variants of MEGA software were used?
-line 466 “members from bamboo, rice, and maize using MCScanX software”
There is no reference.
- line 486. “transformed into the Agrobacterium tumefaciens strain”
Species names should be written italic
Round 2
Reviewer 1 Report
Comments and Suggestions for Authors
This study, like many similar studies of a gene or genome family, is based on sequence analysis by software. Further steps include incorporating something from laboratory research to give the study more than virtuality.
I believe that it should be the other way around. Laboratory research should first reveal some phenomenon or peculiarity of something, and then we should go from there and look for the cause.
We can bury ourselves in viral studies on all genes, but it is useless. So, we need a phenomenon first. Then, the authors must come from a specific situation and not repeat what someone has done many times.